# Effectiveness of Advanced Fire Prevention and Emergency Response Training at Nursing Homes

**DOI:** 10.3390/ijerph192013185

**Published:** 2022-10-13

**Authors:** Wan-Ching Li, Jo-Ming Tseng, Hsin-Shu Huang

**Affiliations:** 1Department of Nursing, Ching-Chyuan Hospital, Taichung 42854, Taiwan; 2Department of Safety, Health, and Environmental Engineering, Central Taiwan University of Science and Technology, Taichung 40601, Taiwan; 3Department of Nursing, Central Taiwan University of Science and Technology, Taichung 40601, Taiwan

**Keywords:** nursing home, advanced fire prevention, emergency response training

## Abstract

In long-term care facilities, there are many residents aged 65+ who do not have the ability to seek shelter by themselves in the case of an emergency. Therefore, it is extremely important that nursing home staff be equipped with correct disaster prevention concepts, emergency survival responses, and hazard mitigation measures. The purpose of this study was to discuss the intervention effectiveness of different fire prevention and emergency response trainings at nursing homes, and the relationship and predictivity of awareness to self-efficacy. We used a quasi-experimental research method and recruited staff from two nursing homes through purposive sampling, using a two-team pre- and post-test design to collect results from 41 individuals in the experiment group and 40 individuals in the control group. The research tool was the “Nursing Home Fire Prevention and Emergency Response Awareness and Self-Efficacy Scale”, which was used to compare the effectiveness of advanced and general fire safety training. After receiving improved advanced fire safety training, the total score and the result of the experiment group in fire prevention and emergency response awareness and self-efficacy were both better than those of the control group who had received only general fire safety training (*p* < 0.001); fire prevention and emergency response awareness had a significant and positive correlation with self-efficacy (*r =* 0.601, *p* < 0.001), and awareness was a significant predictor variable of self-efficacy (*β* = 0.601, *p* < 0.001). This study finds that the key to improving learning effectiveness includes adding a fire science concept chapter when creating fire safety training material in order to strengthen basic awareness; fire safety training should comprehensively introduce all related duties and responsibilities for staff fire defense formation, thereby enabling mutual responsive support for the needs of the site. Moreover, becoming familiarized with the knowledge requires the appropriate frequency of training and enhancement of the staff’s awareness of fire prevention and emergency response, which is the most important key to learning effectiveness.

## 1. Introduction

According to the definition provided by the World Health Organization (WHO), the population of Taiwan has officially become an aged society and is progressing towards a super-aged society. In the WHO’s response to aging, it is proposed that all countries be required to establish long-term care systems in order to fulfill the needs of older people [1]. The average life expectancy of all Taiwanese nationals was 81.32 years in 2020. From the three population age groups measured by the aging index statistics, the percentage of those aged 65 years or over was as high as 16.15% in January 2021 [2]. It is clear that long-term care requirements for Taiwanese nationals are increasing each passing day, and various types of long-term care services have sprung from such needs. By the end of 2020, the number of general care and psychiatric care homes had reached 599; an increase of 62 over the 537 homes counted at the end of 2015 [3]. Studies have indicated that in welfare institutions for the elderly, the policy and execution of fire prevention management systems have not been as effective as predicted, and there is a lot of room for improvement [4]. According to the data from Taiwan’s National Fire Agency, between 2012 and June of 2017, there were 14 fires in the country’s facilities, such as nursing homes and welfare institutions for the elderly, causing 27 deaths and 167 injuries [5]. Through collecting and analyzing the data, it was found that the causes of the 14 fires included arson, accidents involving cigarettes, lighters, or candles, and electric fires. Of all the above causes, arson had caused the most severe casualties and is the most difficult factor to grasp. In addition, most residents in long-term care facilities cannot seek shelter by themselves, and they require the help of others in the case of an emergency and are often extremely challenging to move during rescues and evacuations. Therefore, in addition to the functions of fireproof facilities and fire hazard equipment in long-term care facilities that must meet related regulations such as the Building Act and the Fire Services Act, the disaster prevention knowledge and emergency response abilities of all staff in these facilities appear to be of utmost importance. Vigorous training and internalization of the knowledge should be conducted to prevent the predicament of being legal but not logical.

According to Paragraph 2 in Article 16 of the Nursing Personnel Act, the establishment of nursing homes must meet the regulations set in the “List of Standards for Establishing Nursing Facilities”. As indicated in the List, nursing facilities can be classified as general nursing homes and psychiatric nursing homes, with each type regulated by different establishment standard items, including personnel, nursing service equipment, and the design, structure, and equipment of the building. For general nursing homes, the regulation requires the following: the total number of nursing staff must meet the minimum of one nursing staff member per fifteen beds, and there must be a nursing staff member on duty at any time of the day; one or more nursing aides must be available every five beds, and at any time, the total number of nursing staff and nursing aides to the number of residents must not be lower than one to fifteen; manpower must be suitably increased to accommodate the duties carried out in each work shift. According to Article 15 of the Nursing Personnel Act, the targets of service of nursing homes are patients with chronic illnesses who require long-term care and patients who are discharged from a hospital but continue to require care. The nursing homes are supervised and managed by health authorities and provide various types of nursing care, regular doctor treatments, physiotherapy and occupational therapy activities, nutritional evaluations, and daily care. Other than establishing personnel and equipment based on the “List of Standards for Establishing Nursing Facilities”, the nursing homes must accept supervision and inspection through regular evaluations. According to the study of the first national nursing home evaluation in Taiwan in 2009, the competent authority should exert enhanced counseling to nursing homes to improve in four major aspects: healthcare services, personnel management, operation management, and environment safety [6]. The study showed that personnel education and training account for two of the four key factors for fire prevention and evacuation safety in nursing homes, therefore, it is extremely important to strengthen the emergency response ability of staff and regularly handle drills [7]. The study suggested that the staff of long-term care institutions should take into account the implementation of education and training and fire safety equipment when implementing emergency response to fires [8]. The “Fire Safety Management Guide 2.0 for General Nursing Homes” amended by the Ministry of Health and Welfare in June 2018 emphasizes that through “Increased Fire Hazard Self-Management”, “Fire Hazard Identification and Communication”, and “Simulation and Situational Exercise”, consensus can be formed and the goal of self-managed hazard prevention can be achieved [9]. Such material can be provided to the nursing homes to establish fire prevention and emergency response protocols and trainings. Studies indicate that there is a great gap between the chronic medical care institutions within the country and the evaluation standards of the Joint Commission on Accreditation of Healthcare Organizations (JCAHO) regarding emergency response planning, hazard prevention and mitigation measures, personnel training, and manager operation ideologies. It has been proposed that if staff in the institutions know proper hazard prevention and response concepts, they can easily formulate suitable response measures, and thus, thir response abilities can be enhanced rapidly [10]. With better fundamental awareness of hazard response, the level of emergency response ability will also be improved [11]. Thus, the focus of fire prevention and emergency response training at nursing homes should be on the instillation of concepts and awareness. The current practices at general nursing homes on fire prevention and emergency response training are mostly drafted by the institutions’ manager or fire prevention manager, who incorporate these practices into one of the units in staff trainings according to the required contents indicated in the evaluation standard. The material is usually brief and promotional, and more of a traditional routine training; the fire extinguishing and evacuation process is explained and advised, but there is no emphasis on awareness or fire hazard response [11]. The study findings indicated that to gain awareness and understanding of fires, fire science concepts must be incorporated, including the characteristics of fire hazards, the fire triangle, types of ignition sources, the progression of fires, methods of extinguishing fires, and the dangers of fires. The key factor that affects the success of firefighting and hazard reduction is the staff’s level of understanding and grasp of the progression of fires [12]. Thus, the advanced fire safety training material drafted in this study incorporates a unit on fire science concepts in addition to the Fire Safety Management Guide 2.0 for General Nursing Homes by the Ministry of Health and Welfare.

According to previous studies, different intervention methods of training can increase the effectiveness of students’ awareness of subjects. For students with lower awareness, personalized or adjusted training content can be administered, and the students’ personal background should also be considered when executing training intervention [13]. The researchers found that involvement in the training has a significant positive correlation with training effectiveness. The method of evaluating training effectiveness has been widely studied by various scholars from different angles, including the evaluation of the trainee’s result level and the evaluation of the contribution to enterprise management results [14]. The study proposed that result levels be graded on the performance index based on the effects produced through the training, the performance index of each result level has different evaluation methods, including the usage of questionnaires, interviews, reflection reports, structured tests, and technical operations [15]. The researchers believed that, through training, companies can assist staff in obtaining knowledge and skills, summing the viewpoints of the above scholars, this study used questionnaire scales to evaluate fire hazard training effectiveness [16].

The researchers proposed that to achieve good learning effectiveness, one needs to have a full grasp of the trainees’ basic characteristics such as age, level of education, and work seniority [17]. Research has found that communication skills are one of the stressors for foreign nursing aides and may impact training effectiveness [18]. The study indicated that individuals older in age and with a high level of education have higher self-efficacy, meaning that training effectiveness is relatively better [19]. The study indicated that different types of professionality and the length of working years lead to significant differences in the knowledge retained after training. Based on the above, this study incorporated nationality, age, level of education, work seniority, and job position into the variables [20].

Thus, the goals of this study were to (1) discuss the intervention effectiveness of different fire prevention and emergency response trainings at nursing homes, and (2) understand the correlation and predictability of nursing home staff’s self-efficacy in and awareness of fire prevention and emergency response. It is hoped that the study results can provide directions for trainings in long-term care facilities and become a reference for standards and regulations set in evaluations. 

## 2. Materials and Methods

### 2.1. Study Design and Participants

Through a quasi-experimental study and purposive sampling, data from two affiliated nursing homes of private hospitals registered with the Taichung City Government were collected, and a research sample was retrieved for data analysis. The sampling criteria were as follows: 1. Full time employees of all job positions within the study establishments, meaning the staff members who work regularly at the nursing homes, including nursing staff, Taiwanese nursing aides, foreign nursing aides, human resources, and finance or administrative staff; 2. Employed at the current institution for over 3 months; 3. Consent to participate in this study. A pretest–post-test design with two groups was utilized, with the control group trained with general fire hazard training, and the experimental group accepting intervention through advanced fire hazard training. The “Nursing Home Fire Prevention and Emergency Response Awareness and Self-Efficacy Scale” was created to compare and analyze the self-efficacy and fire prevention and response emergency awareness of the staff of the two nursing homes. G power 3.0 software estimated the required number of samples to be 82. This study had 41 subjects in the advanced fire safety training, and 40 subjects in the general fire safety training, with a total 81 subjects.

### 2.2. Measurement

The “Nursing Home Fire Prevention and Emergency Response Awareness and Self-Efficacy Scale” was created as the measuring tool for this study, with reference to the key points in the “Fire Safety Management Guide 2.0 for General Nursing Homes” published by the Ministry of Health and Welfare. The content includes three main areas: “Personal Background Profile of Nursing Home Staff”, “Awareness of Nursing Home Staff Towards Fire Prevention and Emergency Response”, and “Self-Efficacy of Nursing Home Staff Towards Fire Prevention and Emergency Response”. The Awareness and Self-Efficacy Scale underwent an expert validity check, grading the importance, appropriateness, and text clarity of each question in the scale, which had a total CVI of 0.95. The internal consistency method was used to check reliability, and Cronbach’s α of the internal consistency was 0.954.

#### 2.2.1. Personal Background Profile of Nursing Home Staff

This section is designed based on a literature review and the variables that would like to be discussed; the items include nationality, age, level of education, job position, and working years in long-term care.

#### 2.2.2. Awareness of Nursing Home Staff towards Fire Prevention and Emergency Response

The purpose of this section is to investigate the nursing home staff’s awareness of fire prevention and emergency response, including three main topics of “Fire Safety Equipment Awareness”, “Fire Prevention Awareness”, and “Awareness of Emergency Response Measures in Case of Fire”. 

#### 2.2.3. Self-Efficacy of Nursing Home Staff towards Fire Prevention and Emergency Response

The researchers were interested in investigating the self-efficacy of nursing home staff in fire prevention and emergency response, including three main topics of “Fire Safety Equipment Self-Efficacy”, “Fire Prevention Self-Efficacy”, and “Self-Efficacy towards Emergency Response Measures in Case of Fire”. Five fire-safety-related experts in the country were invited to grade the importance, appropriateness, and text clarity of each variable’s contents in the scale through a three-point scale. The validity CVI used medium standard as the determination standard, and the appropriateness and text clarity of the variables in the scale were revised according to the expert recommendation. The final number of questions in the scale remained at 76, and the total CVI was 0.95. The internal consistency method was used to check the reliability, using 30 staff members in similar hospital-affiliated nursing homes as pretest subjects. The awareness scale used yes–no–unknown questions as a scoring standard, with the answers being “yes”, “no”, or “don’t know”. Right answers were granted 1 point, and wrong and “don’t know” answers were granted no points. The self-efficacy scale used multiple choices, utilizing a 5-point Likert scale for the scoring, with 1 being very unconfident, 2 being not confident, 3 being neutral, 4 being confident, and 5 being very confident. Calculation was conducted separately through the question difficulty and degree of discrimination, and the final number of questions in the scale was reduced from 76 to 62 questions. The pretest’s total internal consistency Cronbach α was 0.954, indicating that the “Nursing Home Fire Prevention and Emergency Response Awareness, and Self-Efficacy Scale” can stably reflect the nursing home staff’s awareness and self-efficacy towards fire prevention and emergency response. 

### 2.3. Ethical Considerations

This study conducted data collection after receiving approval from the Central Regional Research Ethics Committee of China Medical University (IRB No. CRREC-109-079). Prior to the test, the researchers explained the research objectives, process, and rights to the test subjects. After gaining the test subjects’ voluntary consent, the method for filling in the questionnaire was explained, and the subjects were informed that all of their data would be anonymous and processed by numbering. If the test subjects experienced any physical or psychological discomfort, they had the right to exit or terminate the research at any time, and they were assured that their involvement and questionnaire results would not impact their assessments or any rights within the institution. The research subjects could contact the researchers at any time if they had questions, and it was the researchers’ obligation and responsibility to protect and respect the subjects’ privacy to ensure their personal data are absolutely classified. All the collected data were used for academic research analysis only, with absolutely no exposure to the public, clear adherence to research ethics, and protection of the research subjects’ right to privacy and personal information. 

### 2.4. Data Collection and Analysis

The data of this study were collected from two affiliated nursing homes of private hospitals registered with the Taichung City Government, and data collection only commenced after obtaining approval from the two institutions’ managers and passing the review of the institutional review board. The researchers of the study hand-delivered the “Nursing Home Fire Prevention and Emergency Response Awareness and Self-Efficacy Scale” to the two nursing homes. To ensure completion and correctness when filling in the questionnaire, a face-to-face oral explanation was conducted to explain the goal of the study and the key points when filling in. The two groups completed the pretest scale, and the “advanced fire safety training” was implemented in the experimental group, while the “general fire safety training” was implemented in the control group. A post-test was conducted after two weeks, with a 100% recovery rate of the scale. 

The collected scale results were numbered and entered into a computer, using SPSS for Windows 20.0 software to conduct data statistics analysis. The frequency distribution was used for description statistics, percentage descriptions were used for basic profile data, the average and standard deviation were used to describe pretest/post-test scores and training-related effectiveness, and, finally, the chi-squared test was used to inspect the basic profile differences between the test subjects. Inferential statistics were checked using an independent-sample t-test, a paired-sample t-test, and analysis of covariance to analyze the differences in fire prevention and emergency response awareness and self-efficacy between staff members with different basic profiles. The Pearson correlation coefficient was used to investigate the correlation of self-efficacy against fire prevention and emergency response awareness in nursing home staff, and then, simple regression analysis was used to inspect the predictivity of awareness to self-efficacy. The above study variables set a *p* value of *p* < 0.05.

## 3. Results

### 3.1. Participants’ Demographics 

With regard to the nationality distribution of the nursing home staff, the majority of both the experimental group and control group were Taiwanese nationals, at 27 (66%) and 25 people (63%), respectively. In terms of the level of education distribution, the experimental group mostly had college education and above, at 25 people (61%), while the control group mostly had high school education or below, at 22 people (55%). Regarding the job position distribution, both the experimental and control groups had more nursing aides, at 26 (63%) and 27 people (68%), respectively. In regard to the age distribution, both the experimental and control groups were mostly 45 years and under at 33 (80%) and 27 people (68%), where the average of the experimental group was 35.63 years old, the standard deviation was 10.62, and the age range was min 22–max 61, while the average of the control group was 41.90 years old, the standard deviation was 8.73, and the age range was min 25–max 62. In relation to work seniority in long-term care, the majority in the experimental group had under 3 years of experience, at 35 people (85%), whereas the control group mostly had over 3 years of experience, at 28 people (70%). The nationality distribution of the subjects in the experimental and control groups had values of χ^2^ = 0.099, *p* = 0.753; the age distribution had values of χ^2^ = 1.778, *p* = 0.182; level of education had values of χ^2^ = 2.075, *p* = 0.150; the job position distribution had values of χ^2^ = 0.149, *p* = 0.699; and work seniority in long-term care had values of χ^2^ = 25.482, *p* < 0.001. In summary, other than the difference in long-term care work seniority, there were no significant differences between the experimental and control groups in their basic profiles, including nationality, age, level of education, and job position, which are presented in Table 1. 

### 3.2. Difference Analysis of Nursing Home Staff’s Awareness of and Self-Efficacy in Fire Prevention and Emergency Response Prior to and after Fire Safety Training Intervention

Prior to the fire safety training intervention, through analysis of covariance, it was found that the homogeneity of variance test had a result of F(1,79) = 3.166, *p* = 0.079, not reaching the significance of 0.05. The variables in the two groups (post-test average) did not have a significant difference in the error variance, suggesting homogeneity; thus, homogeneity of variance was established. When controlling the covariate variable results (pretest average), the between-group effect (through the variable effect) between the experimental and control groups resulted in a significant level of F(1,78) = 130.500, *p <* 0.001, with an effect size of 0.626. This indicates that the independent variable (two groups) had high explanatory power in the post-test, and that there was a significant difference between the groups. Through multiple comparison, it was shown that the experimental group surpassed the control group (Table 2). After the fire safety training intervention, the total average score in the first section of awareness and the second section of self-efficacy was 3.264 (SD = 0.266) for the experimental group, and 2.673 (SD = 0.286) for the control group, *p* < 0.001, indicating a significant difference in the overall performance of the two groups, and that the experimental group performed better than the control group (Table 3). Through a paired t-test, the post-test minus pretest average score of the two groups was analyzed; the result of the paired variable difference shows that the experimental group’s post-test score improved from the pretest 0.850 (SD = 0.462), *p* < 0.001, reaching statistical significance. The control group had an improvement of 0.023 (SD = 0.220), *p* = 0.519, but did not reach a statistically significant difference (Table 4). Through an independent t-test, the post-test minus pretest average score between the two groups was analyzed for difference in effectiveness. It was shown that the respective total scores of the experimental and control groups were 0.850 (SD = 0.462) and 0.023 (SD = 0.220), *p* < 0.001, indicating that the overall performance of the experimental group was better than that of the control group and had a statistically significant difference (Table 5). 

### 3.3. Difference Analysis of Nursing Home Staff with Different Basic Profiles against the Awareness of and Self-Efficacy in Fire Prevention and Emergency Response 

This section presents the improvement extent of the post-test minus pretest average score of the staff members with different profiles to analyze the significance of the results. From the results shown in Table 6, the nationality factor in the awareness section had *p* = 0.031, indicating a significant difference, and that foreign nationals performed better than Taiwanese nationals. However, the self-efficacy section had *p* = 0.562 and the total average score had *p* = 0.842, which did not reach statistical significance. In terms of work seniority in long-term care, the post-test results were the same in the awareness and self-efficacy sections, and the overall average score had *p* < 0.001, showing that long-term care work seniority had a significant difference, and that those with less than 3 years of experience performed better than those with over 3 years of experience. Other factors such as staff age, level of education, and job position did not achieve statistically significant differences whether in the awareness section, self-efficacy section, or total average score. This refutes the hypothesis that nursing home staff’s differing personal backgrounds would lead to significant differences in the training effectiveness of fire prevention and emergency response.

### 3.4. Correlation and Predictivity of Nursing Home Staff’s Awareness of Fire Prevention and Emergency Response to Self-Efficacy

Through Pearson correlation analysis, the correlation between nursing home staff’s awareness of fire prevention and emergency response and self-efficacy had *r* = 0.601 and *p* < 0.001. This high effect size indicates the positive correlation between awareness and self-efficacy; the higher the awareness score, the higher the self-efficacy score (Table 7). Through simple regression analysis, using the nursing home staff’s awareness of fire prevention and emergency response as a predictor and self-efficacy as a criterion variable (dependent variable), the standard regression coefficients were *β* = 0.601, *t* = 6.688, *p* = 0.011, *F*(1,79) = 44.725, and *R*^2^= 0.361. Thus, it is theorized that nursing home staff’s awareness of fire prevention and emergency response is a significant predictor variable of self-efficacy, as it has a high level of effect size for self-efficacy and can explain 36.1% of the variability (Table 8).

## 4. Discussion

Through the analysis results, it was shown that improved advanced fire safety training surpasses general fire safety training, and that staff members in the experimental group who received the advanced fire safety training performed better in the awareness and self-efficacy sections than the control group who received the general fire safety training. The study results are similar to the other study of “A Study of the Relationship of Employee Self-Efficacy, Learning Strategy, and E-Learning Effectiveness”, which states that when employees’ self-efficacy is high, their learning satisfaction is also higher, with better learning effectiveness [21]. It was also found that after receiving training, the trainees’ knowledge and self-efficacy were both increased when compared with the score before receiving training [22]. Regarding the differences in nursing home staff with different basic profiles and their fire prevention and emergency response awareness, self-efficacy, and total scores, it was found that staff who are Taiwanese nationals performed significantly better than foreign nationals in awareness, self-efficacy, and total score prior to the fire safety training intervention. This study result showed that in residential long-term care facilities, staff who are Taiwanese nationals have a better performance in fire management awareness, attitude, and behavior compared to foreign nationals [23]. With the fire safety training intervention, although the self-efficacy and total score were still high in Taiwanese nationals compared to foreign nationals, and reached a significant difference, there was no significant difference between the two in the awareness section. In the effectiveness analysis, foreign nationals’ performance in awareness was better than that of Taiwanese nationals and reached a significant difference, although there was no significant difference in self-efficacy and the total score. It is evident that after receiving the training, foreign nationals’ improvement in awareness was greater than that of Taiwanese nationals, and their self-efficacy and total score also reached the same levels as those of Taiwanese nationals. This indicates that foreign staff indeed require fire safety training, and that after training, whether in awareness, self-efficacy, or the total average, they can reach the same level as Taiwanese nationals. This signifies that fire safety training has good effectiveness and is more important for foreign staff members. The age groups of 45 years and below and 46 years and above had no statistical difference in the total average score, whether it was prior to or after the training intervention, or in the effectiveness analysis. The study result corresponds to the result of this study [13] but is different from that of the other study, which noted that the older the age, the higher the self-efficacy and training effectiveness [19]. In terms of the level of education, for college and above or high school and below, there was no significant difference in the effectiveness analysis. This result is similar to the result from the study [24]. As for the job position factor, the effectiveness analysis total average of nursing aides and medical administration staff did not reach a significant difference, which is different from the results of the study [20]. The presumed reason is the difference in fire safety education from medical professional training. Through effectiveness analysis, it was found that those with long-term care work seniority of under 3 years performed better than those with work seniority of over 3 years, and reached a statistically significant difference. This finding is different from the findings of the study [13], but similar to the findings of the other study [20]. It was found that, prior to the training, the knowledge score was not related to work seniority. However, through training, the knowledge score increased for each seniority level, but those who had 6–10 years of work experience still scored lower than the other groups, indicating that better performance is not linked to higher work seniority [20]. Through Pearson correlation analysis, it was found that the nursing home staff’s awareness of fire prevention and emergency response had a significant positive correlation with self-efficacy, and that when the awareness score is higher, self-efficacy is also higher. This result corresponds to that of the study and indicates that the better the fire response basic awareness, the better the emergency response ability [11]. The study noted that when staff’s self-efficacy is high, their learning awareness effectiveness is also higher [21]. Through the simple regression analysis results, it was found that there was significant predictivity of the nursing home staff’s awareness of fire prevention and emergency response to self-efficacy. This result is consistent with the results of the study, showing that when a long-term care facility’s staff perform better at fire prevention management, they will have a more confident attitude and behavior [23]. This result is similar to the results of the study [25] and the other study [26], which noted that belief in self-efficacy and training effectiveness are positively correlated.

The research subjects of this study were limited to two nursing homes in Taichung City and are of the same hierarchy. The sample does not cover the entirety of Taichung City or the entire country. It is recommended that future researchers expand the data collection scope in order to investigate the training effectiveness of fire prevention and emergency response in long-term care facilities of different hierarchies and scales. Additionally, the main staff members that provide care in long-term care facilities are nursing aides, among which foreign nationals are abundant. Thus, the prior creation of a questionnaire, the pretest–post-test explanation, and the training intervention were all highly challenging for the researchers. It is recommended that future researchers try to minimize the gap between languages and text communication.

## 5. Conclusions

Through multiple verifications, this study found that an improved advanced fire safety training method and curriculum surpassed traditional fire safety training (Appendix A). Although fire safety knowledge and material can be found on the websites of the National Fire Agency, the Ministry of the Interior, and the Ministry of Health and Welfare, the contents tend to be general or mainly written documents. It is recommended that government and health agencies design custom materials for different types of long-term care facilities, including a fire science concept unit to enhance the staff’s basic awareness of fire science. This can also be provided to trainers for all long-term care facilities for download and utilization. Additionally, the fire safety trainers in long-term care facilities should comprehensively instruct staff members regarding all fire-related duties and responsibilities to enable them to respond and support the needs on site. This is to prevent confusion and an inability to act when the scenario or personnel changes. Moreover, becoming familiar with the knowledge obtained requires an appropriate frequency of training. The study result indicates that the staff member profile is not the main factor that impacts training effectiveness, meaning that each staff member may start at a different level, but through correct and effective training materials and methods, the effectiveness of fire safety training can still be improved. This study also found that the higher the performance in awareness, the higher the self-efficacy and the better the grasp on incident response. Therefore, the enhancement of staff awareness of fire prevention and emergency response by their managers or trainers should be a key focus for increasing learning effectiveness. 

## Figures and Tables

**Table 1 ijerph-19-13185-t001:** Participants’ basic profile distribution and difference comparison (n = 81).

Variables	All (n = 81)	Experimental Group (n = 41)	Control Group (n = 40)	χ^2^ Value	*p* Value
	n (%)	n (%)	n (%)		
Nationality				0.099	0.753
Taiwanese	52 (64)	27 (66)	25 (63)		
Foreign	29 (36)	14 (34)	15 (38)		
Age				1.778	0.182
45 Years and Below	46 (57)	33 (80)	27 (68)		
46 Years and Above	35 (43)	8 (20)	13 (33)		
Level of Education				2.075	0.150
High School and Below	38 (47)	16 (39)	22 (55)		
College and Above	43 (53)	25 (61)	18 (45)		
Job Position				0.149	0.699
Nursing Aide	53 (65)	26 (63)	27 (68)		
Medical Administration	28 (35)	15 (37)	13 (33)		
Work Seniority in Long-Term Care				25.482	<0.001 ***
Below 3 Years	47 (58)	35 (85)	12 (30)		
Above 3 Years	34 (42)	6 (15)	28 (70)		

Note: *** *p* < 0.001.

**Table 2 ijerph-19-13185-t002:** Analysis of covariance on research subject training effectiveness (N = 81).

Source of Variable	Type I Sum of Square	Degree of Freedom	Average Sum of Square	*F*	*p*	Effect Size	Multiple Comparison
Pretest	0.053	1	0.053	0.854	0.358	0.011	Experimental Group > Control Group
Two Groups	8.16	1	8.160	130.5	<0.001 ***	0.626	
Within-Group (Deviation)	4.848	78	0.063				
All	728.414	81					

Note: *** *p* < 0.001.

**Table 3 ijerph-19-13185-t003:** Comparison of research subjects’ awareness of and self-efficacy in fire prevention and emergency response after fire safety training intervention (N = 81).

Item	Experimental Group (n = 41)		Control Group (n = 40)		*t*	*p*	95%	CI
	*M*	*SD*	*M*	*SD*			*LL*	*UL*
Awareness								
Fire safety equipment awareness	0.959	0.077	0.588	0.160	13.205	<0.001 **	0.315	0.427
Fire prevention awareness	0.963	0.105	0.656	0.257	6.994	<0.001 **	0.219	0.395
Emergency response awareness in case of fire	0.930	0.093	0.569	0.144	13.363	<0.001 **	0.307	0.415
Awareness Total Average	0.949	0.068	0.593	0.138	14.641	<0.001 **	0.307	0.405
Self-Efficacy								
Fire safety equipment self-efficacy	4.473	0.437	3.929	0.377	6.000	<0.001 **	0.363	0.724
Fire prevention self-efficacy	4.580	0.430	3.782	0.478	7.908	<0.001 **	0.598	1.000
Emergency response self-efficacy in case of fire	4.551	0.459	3.738	0.473	7.853	<0.001 **	0.607	1.019
Self-Efficacy Total Average	4.537	0.410	3.816	0.415	7.857	<0.001 **	0.538	0.903
Total Average	3.264	0.266	2.673	0.286	9.625	<0.001 **	0.469	0.713

Note 1: ** *p* < 0.01; 2: The *t* value is the independent-sample *t*-test result.

**Table 4 ijerph-19-13185-t004:** Effectiveness comparison of the pretest–post-test of the two groups (N = 81).

Item	Pretest		Post-Test		Post-Test MinusPretest		*t*	*p*	95%	CI
	*M*	SD	*M*	*SD*	*M*	*SD*			*LL*	*UL*
Awareness										
Fire safety equipment awareness										
Experimental Group	0.512	0.154	0.959	0.077	0.447	0.176	16.219	<0.001 ***	0.390	0.502
Control Group	0.558	0.126	0.588	0.160	0.030	0.124	1.525	0.135	−0.010	0.070
Fire prevention awareness										
Experimental Group	0.500	0.358	0.963	0.105	0.463	0.373	7.952	<0.001 ***	0.346	0.581
Control Group	0.469	0.331	0.656	0.258	0.187	0.276	4.298	<0.001 ***	0.099	0.276
Emergency response awareness in case of fire										
Experimental Group	0.476	0.177	0.930	0.093	0.454	0.207	14.063	<0.001 ***	0.389	0.520
Control Group	0.447	0.147	0.569	0.144	0.122	0.161	4.777	<0.001 ***	0.070	0.173
Awareness total average										
Experimental Group	0.497	0.158	0.949	0.068	0.452	0.178	16.238	<0.001 ***	0.396	0.509
Control Group	0.501	0.141	0.593	0.138	0.092	0.115	5.049	<0.001 ***	0.055	0.129
Self-Efficacy										
Fire safety equipment self-efficacy										
Experimental Group	3.447	0.704	4.473	0.437	1.026	0.680	9.662	<0.001 ***	0.812	1.241
Control Group	3.919	0.431	3.929	0.377	0.010	0.350	0.174	0.863	−0.102	0.122
Fire prevention self-efficacy										
Experimental Group	3.623	0.728	4.581	0.430	0.958	0.743	8.253	<0.001 ***	0.723	1.192
Control Group	3.828	0.504	3.782	0.478	−0.046	0.417	−0.708	0.483	−0.180	0.087
Emergency response self-efficacy in case of fire										
Experimental Group	3.297	0.741	4.551	0.459	1.254	0.816	9.842	<0.001 ***	0.997	1.512
Control Group	3.742	0.535	3.738	0.473	−0.004	0.410	−0.064	0.949	−0.135	0.127
Self-efficacy total average										
Experimental Group	3.468	0.674	4.537	0.410	1.069	0.693	9.872	<0.001 ***	0.850	1.288
Control Group	3.832	0.433	3.816	0.415	−0.016	0.336	−0.294	0.770	−0.123	0.092
Total Average										
Experimental Group	2.414	0.450	3.264	0.266	0.850	0.462	11.780	<0.001 ***	0.704	0.996
Control Group	2.650	0.304	2.673	0.286	0.023	0.220	0.650	0.519	−0.048	0.093

Note 1: *** *p* < 0.001; 2: The *t* value is the paired-sample *t*-test result.

**Table 5 ijerph-19-13185-t005:** Difference analysis of the two groups’ effectiveness in the pretest and post-test (N = 81).

Item	Experimental Group (n = 41)		Control Group (n = 40)		*t*	*p*	95%	CI
	*M*	*SD*	*M*	*SD*			*LL*	*UL*
Awareness								
Fire safety equipment awareness	0.446	0.176	0.030	0.124	12.256	<0.001 **	0.349	0.484
Fire prevention awareness	0.463	0.373	0.188	0.276	3.790	<0.001 **	0.131	0.421
Emergency response awareness in case of fire	0.454	0.207	0.122	0.161	8.051	<0.001 **	0.250	0.415
Awareness Total Average	0.452	0.178	0.092	0.115	10.822	<0.001 **	0.294	0.427
Self-Efficacy								
Fire safety equipment self-efficacy	1.026	0.680	0.010	0.350	8.487	<0.001 **	0.777	1.256
Fire prevention self-efficacy	0.958	0.743	−0.047	0.417	7.526	<0.001 **	0.738	1.271
Emergency response self-efficacy in case of fire	1.254	0.816	−0.004	0.410	8.800	<0.001 **	0.972	1.544
Self-Efficacy Total Average	1.069	0.693	−0.016	0.336	8.993	<0.001 **	0.843	1.326
Total Average	0.850	0.462	0.023	0.220	10.332	<0.001 **	0.667	0.988

Note 1: ** *p* < 0.01; 2: The *t* value is the independent-sample t-test result.

**Table 6 ijerph-19-13185-t006:** Analysis of staff members with different basic profiles and their fire safety training effectiveness (N = 81).

Items	Awareness			Self-Efficacy			Total Average		
	*M*	*SD*	*p*	*M*	*SD*	*p*	*M*	*SD*	*p*
Nationality									
Taiwanese	0.233	0.221	0.031 *	0.571	0.785	0.562	0.451	0.556	0.842
Foreign	0.350	0.245		0.466	0.751		0.425	0.551	
Age									
45 Years and Below	0.299	0.241	0.108	0.581	0.764	0.351	0.481	0.546	0.279
46 Years and Above	0.203	0.207		0.398	0.789		0.329	0.562	
Level of Education									
High School and Below	0.269	0.249	0.851	0.501	0.775	0.722	0.419	0.569	0.727
College and Above	0.279	0.225		0.562	0.774		0.462	0.541	
Job Position									
Nursing Aide	0.306	0.250	0.072	0.538	0.849	0.938	0.456	0.614	0.725
Medical Administration	0.214	0.194		0.525	0.606		0.415	0.421	
Work Seniority in Long-Term Care									
Below 3 Years	0.362	0.233	<0.001 ***	0.797	0.822	<0.001 ***	0.643	0.579	<0.001 ***
Above 3 Years	0.154	0.180		0.168	0.509		0.163	0.362	<0.001 ***

Note: * *p* < 0.05, *** *p* < 0.001.

**Table 7 ijerph-19-13185-t007:** Correlation analysis between awareness of fire prevention and emergency response and self-efficacy (N = 81).

	Pretest		Post-test		Post-Test Minus Pretest	
	Awareness	Self-Efficacy	Awareness	Self-Efficacy	Awareness	Self-Efficacy
Pearson correlation	1	0.281	1	0.641	1	0.601
Significance (two-tailed)		0.011 *		<0.001 ***		<0.001 ***
Number	81	81	81	81	81	81

Note: * *p* < 0.05, *** *p* < 0.001.

**Table 8 ijerph-19-13185-t008:** Predictivity analysis between awareness of fire prevention and emergency response and self-efficacy (N = 81).

	Standardized Coefficient	*R*^2^ Value	*F* Test Value	*t* Value	*p* Value	95%	CI
	Beta					*LL*	*UL*
Dependent variable							
Self-efficacy							
Predictor variable							
Awareness	0.601	0.361	44.725	6.688	<0.001 ***	1.384	2.556

Note: *** *p* < 0.001.

## Data Availability

All the relevant datasets in this study are described in the manuscript.

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
