# Peer review of "Effectiveness of Advanced Fire Prevention and Emergency Response Training at Nursing Homes"

_ijerph, 2022, doi:10.3390/ijerph192013185_

Round 1

Reviewer 1 Report (Previous Reviewer 2)

The manuscript (MS) analyses the difference in advanced fire prevention and emergency response training at nursing homes compared to the traditional training regime. The study concludes that introducing fire science etc. in the training program greatly improved the knowledge of the participants regardless of their education level.

Title: The title of the MS is clear and descriptive

Abstract:

Check the IJERPH template regarding Line 11, 12, 15/16, 18 and 29.

Line 35: Here you conclude about the frequency of training. But as far as I could see, this is only mentioned in the conclusion, and then introduced in the Abstract. See comments for Section 5.

1. Introduction

Line 115: Follow the template regarding referencing, i.e., no parenthesis with year after the author name(s). Please check carefully throughout the MS. Numerous of these corrections are required. (If you need the year to be stated, write, e.g., "In 2012, Cai at al. [6] presented the first national nursing home evaluation in Taiwan in 2009." An advice: Search for "(" throughout the study and remove the "(year)" and comply fully to the IJERPH template, Line 118, 121, etc. etc.

2. Materials and Methods

Line 459: This is a very long sentence. Split it in separate sentences. (Check this throughout as there are several 50+ words sentences that need to be split in two or three sentences for improved readability, e.g., Line 662 to 666, etc.)

3. Results

Line 673: In the similar size groups, 25 (experimental group) vs 18 (control group) have college and above level of education. Is that not a significant difference? 25 is nearly 40% more than 18. This needs to be clarified in the results, and discussed in section 4.

Line 702: After this 4th round of review, please follow the template, i.e., "3.2. Difference Analysis of Nursing Home …"

Line 784: See comment for Line 702.

Line 821: See comment for Line 702.

4. Discussion

Line 841: A very long sentence…, but I suppose that you already found this one and the other long sentences earlier in the MS when searching through the text?

And the parenthesis in Line 845 has already been deleted?

Page 11: Contains at least one very long sentence and 10+ "(year)" that needs to be deleted, while, e.g., the reference to Bi et al. [24] in Line 900 is correct.

5. Conclusions

Line 1133: Here you introduce a conclusion about the appropriate frequency of training. As far as I can see, this has not been mentioned before. The conclusion should not add new information. Delete it, or discuss this elsewhere in the MS with reference to work that support this (rather obvious) conclusion, so it can be stay in this section – and in the Abstract.

Besides these comments, the MS has been significantly improved from the first submission, i.e., as ijerph-1759100-peer-review-v1.pdf.

Author Response

Response to Reviewer:

Accept your valuable suggestions about the English language and style are fine/minor spell check is required. The full manuscript is professionally checked for the English language and style. Please review the attached document, thank you!

Sincerely,

Hsin-Shu Huang

Department of Nursing, Central Taiwan University of Science and Technology

Mobile: 886-970385717

Fax: +886-4-22391647#7385

E-mail: hshuang@ctust.edu.tw, hsinshu888888@gmail.com

Address: 666 Pu-tzu Road Taichung 40601, Taiwan

This manuscript is a resubmission of an earlier submission. The following is a list of the peer review reports and author responses from that submission.

Round 1

Reviewer 1 Report

Abstract section

Purpose: Discuss the intervention effectiveness of different fire prevention and emergency response 16 trainings at nursing homes and the relationship and predictivity of awareness to self-

Suggestions Better as:

Purpose: To discuss effectiveness of different intervention on fire prevention and emergency response ……

Method: Recruit staff from two nursing homes through purposive sampling, using a two-team pre- and post-test design to collect results from 41 individuals in the experiment group and 40 individuals in the control group.

Suggestions from reviewer:

First, it is better to state clearly the research method adopted. For example, what type of data was collected by the two-team pre- and post-test design? Quantitative, Qualitative or mixed research methods data? I can tell that your study adopted quantitative research method based on data and result presented in you study. But, you must clearly state the research method adopted. Besides, you must state how you analysed your data? For example, the data collected were analysed using SPSS, with correlation analysis or regress analysis, performed to determine the relationship between x and y.

Result: After receiving improved advanced fire safety training, the total score and the result of the experiment group on fire prevention and emergency response awareness and self-efficacy had 24 both performed better than the control group who received general fire safety training (p 25 < .001); fire prevention and emergency response awareness has significant and positive

Suggestions from reviewer:

How did you arrive at the total score and the result of the experiment group on fire prevention and emergency response? Please refer to my earlier suggestion above about method of data analysis

This study finds that 28 the key to improving learning effectiveness includes adding fire science concept chapter 29 when creating fire safety training material in order to strengthen basic awareness; fire 30 safety training should comprehensively introduce all related duty ….

Suggestions from reviewer:

Incorrect sentence. Besides, it is better to write in plural rather singular tenses when making suggestions to e.g. materials

Better as:

Findings from the study reveal 28 keys to improving learning … when creating fire safety training materials ….

Introduction to the study

Introduction section is fine. However, it is better to clearly state the rationale for the study or state the key variables mentioned in the study title i.e. “Advanced fire prevention”, “emergency response training” and “nursing homes”

 But there is a need to improve use grammar and syntax. I can tell that English is not the first language of the authors, but there is a need to improve writing style. For example:

Paragraph 1 continuous numbering 39 -  “According to the definition by the World Health Organization (WHO) …”

Paragraph 1 continuous numbering 71 – “According to Paragraph 2 in Article 16 of the Nursing Personnel Act …”

It is better to use variety of words and phrases when starting paragraph. It is old fashion to use say “According to …” in paragraph 1 and use “According to “again start paragraph 2.

Literature Review

The study literature is scanty. There is significant literature in fire prevention, emergency training / response and nursing homes. It is necessary to robustly develop the study literature review.

Research Method

There is a need for the authors to state type of data was collected by the two-team pre- and post-test design? Clearly identify research method adopted that is Quantitative, Qualitative or mixed research methods data? I can tell that your study adopted quantitative research method based on data and result presented in you study. But, you must clearly state the research method adopted. Explanation of how the study data was analysed is not clear. For example, the data collected were analysed using SPSS, with correlation analysis or regress analysis, performed to determine the relationship between x and y.

 Conclusion is fair and reflective of the study fine. 

Finally, Syntax problem commonplace in the study. There study need thorough proofreading

Author Response

Response to Reviewer 1 Comments

Point 1: Suggestions from the reviewer:

First, it is better to state clearly the research method adopted. For example, what type of data was collected by the two-team pre-and post-test design? Quantitative, Qualitative, or mixed research methods data? I can tell that your study adopted a quantitative research method based on data and results presented in your study. But, you must clearly state the research method adopted. Besides, you must state how you analyzed your data? For example, the data collected were analyzed using SPSS, with correlation analysis or regress analysis, performed to determine the relationship between x and y.

 Response 1:

  1. In the text 2.1 Study Design and Participant (lines 149-168), it is explained that this study uses a quasi-experimental study design and purposive sampling, so this study is a quantitative study and a non-qualitative study. It also explains the conditions of the study subjects.

Line149-168:

2.1. Study Design and Participant

Through quasi-experimental study and purposive sampling, data from two affiliated nursing homes of private hospitals registered with the Taichung City Government was collected and a research sample was retrieved for data analysis. The sampling criteria were: 1. Full-time employees of all job positions within the study establishments, meaning the staff members who work regularly at the nursing homes, including nursing staff, Taiwanese nursing aides, foreign nursing aides, human resources, finance or administrative staff, etc. 2. Employed at the current institution for over 3 months. 3. Consent to join this study. Pretest-posttest design in two groups was utilized, with the control group trained with general fire hazard training, and the experimental group accepting intervention through advanced fire hazard training. The “Nursing Home Fire Prevention and Emergency Response Awareness and Self-Efficacy Scale” was created to compare and analyze the self-efficacy and the fire prevention and response emergency awareness of the staff of the two nursing homes. The Awareness and Self-Efficacy Scale underwent an expert validity check, grading the importance, appropriateness, and text clarity of each question in the Scale, which had a total CVI of 0.95. The internal consistency method was used to check reliability, and the Cronbach’α of the internal consistency was 0.954. G power 3.0 software had estimated the required number of samples to be 82. This study had 41 subjects in the advanced fire safety training and 40 subjects in the general fire safety training, with a total of 81 subjects.

  1. 2.4 Data analysis (lines 236-247) in the text indicates that this study uses SPSS for Windows 20.0 software to conduct data statistics analysis.

Line236-247:

The collected Scale results were numbered and entered into the computer, using SPSS for Windows 20.0 software to conduct data statistics analysis. Frequency distribution was used on description statistics, percentage description on basic profile data, average, and standard deviation was used to describe pretest/post-test scores and training-related effectiveness, and finally, the chi-squared test was used to inspect the basic profile differences between test subjects. Inferential statistics were checked using an independent sample t-test, paired sample t-test, and analysis of covariance to analyze the differences in fire prevention and emergency response awareness and self-efficacy between staff members with different basic profiles. Pearson correlation coefficient was used to investigate the correlation of self-efficacy against fire prevention and emergency response awareness in nursing home staff, then simple regression analysis was used to inspect the predictivity of awareness to self-efficacy.

 Point 2: Suggestions from the reviewer:

Incorrect sentence. Besides, it is better to write in the plural rather than singular tenses when making suggestions to e.g. materials

Better as:

Findings from the study revealed 28 keys to improving learning … when creating fire safety training materials ….

Response 2:

Thanks to reviewer 1 for your advice. For the sake of perfection, after correcting the typo numbers in Table 4, the full text has been applied for MDPI English major repairing, as shown in the attachment, thank you very much.

Reviewer 2 Report

The manuscript (MS) analyses the difference in advanced fire prevention and emergency response training at nursing homes compared to the traditional training regime. The study concludes that introducing fire science etc. in the training program greatly improved the knowledge of the participants regardless of their education level. Interestingly, the well educated benefitted most when exposed to the improved training program.

A general weakness of the MS is that the contents of the two training regimes are not sufficiently revealed/described. This reviewer suggests that this needs to be elaborated, e.g., in an Appendix, as this would be a) give indications as to what training programs others should develop for good learning and b) more details are necessary for other researchers to repeat the study, and c) possibly develop even better training programs.

Title

The title is representative for the study.

You must follow the Template, i.e., with capital letters throughout: "Effectiveness of Advanced Fire Prevention and Emergency…"

Check for capital letters also in sub-sections, as these are missing in, e.g., 3.2, 3.3, and 3.4.

Abstract

Line 11: follow the Template in this section heading

Line 13: "evacuate" rather than "seek shelter"? Check throughout.

Line 15: consider writing hazard mitigation measures before emergency survival responses as that would chronological order?

The last sentence, commencing at Line 28 and finishing at Line 35 is extremely long. Try to cop it into several sentences to make it more readable.

1. Introduction

Line 58: Consider using the wording evacuate rather than seek shelter

Line 62 (or elsewhere): Define what you mean with disaster prevention concept and include a reference, if possible. See also Line 107-109.

Line 65-70: May I suggest that you move this to the end of the Introduction?

2. Materials and Methods

Line 149: Participants (plural) ?

In Section 2.1, please state whether the hours of training were the same, or that the advances course was extended in the number of hours, etc.

Ans, as required in the start, you need to describe the two training programs better. To not make Section 2.1 too big, may I suggest that you write necessary details, e.g., in an Appendix? Or maybe one Appendix for the traditional training and one for the advanced training program?

Line 178: "The designed through referencing literature reviews and …" ? Rewrite.

Line 183: "… the awareness of … ?

Line 188-191: Much of this has been stated in the introduction to Section 2.2. Is it necessary to repeat it?

Line 193: "… contents in a three-point scale." I.e., short and concise? Check throughout for similar unnecessary writing.

3. Results

Table 4: Fire safety equipment awareness.

Experimental Group   0.512   0.154   0.959 … (Seems OK)

Control Group            0.558   0.126   0.558 … (This does not make a difference of 0.030. A typing error? Check throughout)

Why capital letter in group, i.e., Group? Check throughout.

Line 307: This section presents… I.e., present tense as it always presents.

Line 328: This high effect size indicates … I.e., present tense as it always indicates.

4. Discussion

Given that you add an appendix (or two?) to better describe the training programs content, you may expand the discussion section to better explain the differences in learning outcome.

Line 371, 375, 377, 388: New paragraph(s)?

Line 399: belief (noun) rather than believe (verb)?

5. Conclusions

Line 424: Is the word result redundant? I.e., The study that indicates that staff…

Line 425: consider replacing the word "reason" with parameter or factor, or something similar. Or maybe rewrite the sentence.

Line 427: Consider also rewriting this statement: "… the effectiveness of fire safety training can still be achieved." I suppose that the knowledge level of the participants that can be achieved is the goal.

References

Add DOI’s where available.

Check the Template carefully for each reference. E.g., publication name in italic for 17, 18, 19, …

I look forward to receiving a revised version including valuable details regarding the curriculums of the two training programs, teaching hours, possible practical field training, group, or individual exercises, etc. as well as any other possible advice for improved training programs. That would greatly increase several of the scored Ratings, e.g., the Significance of the study, the Interest to the readers, etc

Author Response

Response to Reviewer 2 Comments

Point 1: Suggestions from the reviewer:

You must follow the Template, i.e., with capital letters throughout: "Effectiveness of Advanced Fire Prevention and Emergency…"

Response 1:

  1. The words that need to be capitalized in the title have been corrected as suggested by the review committee. For example, attach the uploaded file:

Effectiveness of Advanced Fire Prevention and Emergency Response Training at Nursing Homes

Point 2: Suggestions from the reviewer:

Table 4: Fire safety equipment awareness.

Experimental Group   0.512   0.154   0.959 … (Seems OK)

Control Group        0.558   0.126   0.558 … (This does not make a difference of 0.030. A typing error? Check throughout)

Response 2:

Thanks to reviewer 2 for your advice. For the sake of perfection, after correcting the typo numbers in Table 4, the full text has been applied for MDPI English major repairing, as shown in the attachment, thank you very much!

Table 4. Effectiveness comparison of pretest-posttest of two groups (N=81).

Item

Pretest

Posttest

Posttest minus

Pretest

t

P

95%

CI

M

SD

M

SD

M

SD

LL

UL

Awareness

Fire safety equipment awareness

Experimental Group

0.512

0.154

0.959

0.077

0.447

0.176

16.219

<0.001 ***

0.390

0.502

Control Group

0.558

0.126

0.588

0.160

0.030

0.124

1.525

0.135

-0.010

0.070

Fire prevention awareness

Experimental Group

0.500

0.358

0.963

0.105

0.463

0.373

7.952

<0.001 ***

0.346

0.581

Control Group

0.469

0.331

0.656

0.258

0.187

0.276

4.298

<0.001 ***

0.099

0.276

Emergency response awareness in case of fire

Experimental Group

0.476

0.177

0.930

0.093

0.454

0.207

14.063

<0.001 ***

0.389

0.520

Control Group

0.447

0.147

0.569

0.144

0.122

0.161

4.777

<0.001 ***

0.070

0.173

Awareness total average

Experimental Group

0.497

0.158

0.949

0.068

0.452

0.178

16.238

<0.001 ***

0.396

0.509

Control Group

0.501

0.141

0.593

0.138

0.092

0.115

5.049

<0.001 ***

0.055

0.129

Self-Efficacy

Fire safety equipment

self-efficacy

Experimental Group

3.447

0.704

4.473

0.437

1.026

0.680

9.662

<0.001 ***

0.812

1.241

Control Group

3.919

0.431

3.929

0.377

0.010

0.350

0.174

0.863

-0.102

0.122

Fire prevention self-efficacy

Experimental Group

3.623

0.728

4.581

0.430

0.958

0.743

8.253

<0.001 ***

0.723

1.192

Control Group

3.828

0.504

3.782

0.478

-0.046

0.417

-0.708

0.483

-0.180

0.087

Emergency response self-efficacy in case of fire

Experimental Group

3.297

0.741

4.551

0.459

1.254

0.816

9.842

<0.001 ***

0.997

1.512

Control Group

3.742

0.535

3.738

0.473

-0.004

0.410

-0.064

0.949

-0.135

0.127

Self-efficacy total average

Experimental Group

3.468

0.674

4.537

0.410

1.069

0.693

9.872

<0.001 ***

0.850

1.288

Control Group

3.832

0.433

3.816

0.415

-0.016

0.336

-0.294

0.770

-0.123

0.092

Total Average

Experimental Group

2.414

0.450

3.264

0.266

0.850

0.462

11.780

<0.001 ***

0.704

0.996

Control Group

2.650

0.304

2.673

0.286

0.023

0.220

0.650

0.519

-0.048

0.093

Note 1: *p < 0.05, ** p < 0.01, *** p < 0.001.

2: t value is the Paired Samples t-test result.

Point 3: Suggestions from the reviewer:

Add DOI’s where available.

Check the Template carefully for each reference. E.g., publication name in italic for 17, 18, 19, …

Response 3:

  1. Literature 17, 18, 19 The italics of the journals published have been corrected as attached files, and the attached DOI is as follows:
  2. Literature 18 corrected author names as follows:
  3. Literature 19 corrected the number of pages of the published journal as follows:
  4. (17) Liu, Y. Q. The Relationships among Employee's Self Efficacy, Training Needs, and Training Effect. Hsuan Chuang Journal of Social Sciences.2011, 9, 1-20. https://doi.org/10.29592/bgyy.201111.0001
  5. (18) Chu, C. M.; Liu, C. R.; Huang, S. J.; Woung, L. C.; Chen, S. T.; Wu, M. H.; Lin, H. C.; Chen, P. L.; Huang, T. C. The Effectiveness of Community-based Palliative Care Training on Knowledge, Attitude, and Skills for the Terminal Patients. Taipei City Medical Journal. 2015, 12, 86-108. https://doi.org/10.6200/tcmj.2015.12.sp.09
  6. (19) Huang, J. T.; Wang, Y. C.; Li, T. C. A Study of the Relationship of Employee Self-Efficacy, Learning Strategy, and E-Learning Effectiveness. K.A.U.S. Journal of Humanities and Social Science. 2009, 6, 2, 283-306. https://doi. org/10.29888/kuasjhss.200912.0006

 Point 4: Suggestions from the reviewer:

I look forward to receiving a revised version including valuable details regarding the curriculums of the two training programs, teaching hours, possible practical field training, group, or individual exercises, etc. as well as any other possible advice for improved training programs. That would greatly increase several of the scored Ratings, e.g., the Significance of the study, the Interest to the readers, etc.

 Response 4:

Appendix 1. Comparison of General and Advanced Fire Prevention and Emergency Response Training Course Content Design 

General Fire Prevention and Emergency Response Training

Advanced Fire Prevention and Emergency Response Training

PPTs teaching

PPTs may present content

1. Presentation of the internal SOP description of each nursing home

2. Video of news cases

3. Brief description of fire fighting equipment

Re-planning and setting the PPTs teaching content should include the following:

1. Fire Science Concept Unit (10 minutes)

(1) Characteristics of fire

(2) Three elements of combustion

(3) Types of ignition sources

(4) The itinerary of the fire

(5) Methods of extinguishing fire

(6) Hazards of fire

2. Knowledge of fire prevention and emergency response (20 minutes)

(1) Knowledge of fire safety equipment

(2) Knowledge of fire prevention

(3) Cognition of emergency response in case of fire

3. The internal SOP description of each nursing home is presented, and the content is based on each customization can be roughly divided into the following key points (10 minutes)

(1) Scope of application

(2) Grading

(3) Fire notification flow and processing flow

(4) Principles and moving lines of evacuation

4. Video sharing of news cases (5 minutes)

5. Teaching interaction (15 minutes)  The lecturer provides questions about situations that may cause fire and asks students to respond and interact.

Teaching firefighting equipment operation

Demonstration by the teacher, practical operation by the learner Fire extinguishers, fire hydrants, escalators, conveyor belts, etc.

Demonstration by the teacher, practical operation by the learner

Fire extinguishers, fire hydrants, escalators, conveyor belts, etc.

Situational simulation practice Situational drills

If a fire is found in a house or area, assign students to drill

RACE process

Situational drills

If a fire is found in a house or area, assign students to drill

RACE process

Round 2

Reviewer 2 Report

In the first review, I suggested numerous changes. The normal task of the authors is to address each of these shortly in the answer to the reviewer. It should be unnecessary to repat messages such as, e.g., "Check for capital letters also in sub-sections, as these are missing in, e.g., 3.2, 3.3, and 3.4." This means that, e.g., Section 3.2 heading should be written as:

3.2. Difference Analysis of Nursing Home Staff's Awareness …

An othe example: "Add DOI’s where available."

S far as I can see, you have not added any DOIs in the reference list despite the Template and my notification. (Please comply (or refute if necessary), and answer to each point of the reviewers comments separately.)

I also suggest that you mark each change in the manuscript by a colour, e.g., green. (This does not go for the numerous minor language corrections.)

I refer to line numbers in the hide changes mode:

Line 28: Use space between constant/variable and = and between = and the number, i.e., r = 0.601, p < 0.001), … p < 0.001. Carefully check this throughout the manuscript.

Referencing issues:

Line 92: "Cai et al. (2012) studied the first national nursing home evaluation in Taiwan in 2009. Their study indicated that the competent authority should exert enhanced counseling to nursing homes to improve in four major aspects: healthcare services, personnel management, operation management, and environment safety [6]." Since IJERPH uses the bracket system, rewrite to: Cai et al. [6] studied… (Or if it is necessary to state that this was in 2012, e.g., In 2012, Cai et al. [6] studied… Or since they may indeed hav studied it in 2010 or 2011, and published it in 2012, It may be even better to write something like: In 2012, Cai et al. [6] presented a study on…".

Thus, in gerenal, get rid of the parentheses throughout and state the year of the published article, if strictly necessary, in a different way.

The person that helped you out with the language issues has suggested numerous good improvements. (S)he did, however, not realize that Section 2.2.2 and 2.2.3 tells about the form which was created as a measuring tool for the present study.

May I suggest for 2.2.2:

"This section of the measurement tool was designed based on a literature review and the variables that would like to be discussed; the items included nationality, age, level of education, job position, and working years in long-term care."

And for 2.2.2:

"The purpose of this section of the measurement tool was to investigate…"

Line 255: Since you are describing the general Taiwan nursing home staff, the word "the" in front of "nursing home staff" should be deleted, i.e., "With regard to the nationality distribution of nursing home staff, the majority of …"

Line 287-298: Several minor issues with missing space between = and number and a strange "0. 001".

Tables (also commented before): Why some descriptions with capital letters and some with normal letters? E.g., Table 2: "Emergency response self-efficacy in case of fire" vs "Self-Efficacy Total Average"

"Awareness Total Average" seems to be a heading. Why does the heading contain data to the right while "Self-Efficacy" does not, but those below do?

Line 320 and 321: Here, the "p = number" is correctly written while in line 323 it should be written as: "p < 0.001".

Check also Line 337 and 343 and write it correctly. (Search for the term "<" and "=" may help identify these misspellings.)

Line 363: "In the differences in nursing home staff with different basic profiles and their fire pre-vention and emergency response awareness, self-efficacy, and total score, it was found that staff who are Taiwanese nationals performed significantly better than foreign na-tionals in awareness, self-efficacy, and the total score prior to the fire safety training intervention." This tentence is 52 words long, and previoussly commented on from my side. Rewrite it to make sense.

What about something like this, which virtually tells the same story while 15 words shorter:

"Regarding nursing home staff basic profiles, fire prevention and emergency response awareness, self-efficacy, and total score, Taiwanese nationals performed significantly better than foreign nationals in awareness, self-efficacy, and total score prior to the fire safety training intervention."

Besides these issues, the manuscript has been significantly improved.

Author Response

Dear reviewer:

Please check the attached files after the correction. Thank you very much.

Sincerely,

Hsin-Shu Huang

Department of Nursing, Central Taiwan University of Science and Technology

Mobile: 886-970385717

Fax: +886-4-22391647#7385

E-mail: hshuang@ctust.edu.tw, hsinshu888888@gmail.com

Address: 666 Pu-tzu Road Taichung 40601, Taiwan

Round 3

Reviewer 2 Report

In the first review, I suggested numerous changes. Several of these were not answered to in the report to the reviewer. In the second review I noted you about this issue. When receiving the last version, the ony comment was: "Dear reviewer: Please check the attached files after the correction. Thank you very much. Sincerely, …" The file you attached was completely equal to the enclosed manuscript "ijerph-1759100-peer-review-v3.pdf", and gave no new information whatsoever. In both submission v2 and v3 you have simply not followed the "www.mdpi.com/journal/ijerph/instructions":

"All reviewer comments should be responded to in a point-by-point fashion. Where the authors disagree with a reviewer, they must provide a clear response."

Did you ever carefully read (and later on re-read) the instructions for authors?

It is a serious breach of procedure not to respond in a point-by-point fashion, and several issues are therefore not corrected in the version v2, or now in v3.

I suggest that IJERPH rejects this manuscript simply due to the breach of procedure resulting in a manuscript missing many improvement points that should have been completed while leaving the reviewer in darkness. Many of the obvious improvement points were compliant to the IJERPH Template, but still simply ignored or neglected. Quite unprofessional and unpolite.

Author Response

Response to Reviewer 3 Comments

Point 1: The abstract describes the research methodology.

Response 1: The research methods in the abstract describe this study as an quasi experimental research method.

Point 2: The abstract is supplemented with relevant and predicted statistical values.

Response 2: The abstract adds the relevant statistical values r=.601, p< .001 and the predicted statistical values β = .601, p < .001.

Point 3: Add literature review.

Response 3: In 1. Introduction, added Tsai (2021) and Huang et al. research literature (2021): Tsai (2021)

study showed personnel education and training account for two of the four key factors for fire

prevention and evacuation safety in nursing homes, so it is extremely important to strengthen the

emergency response ability of staff and regularly handle drills [7]. Huang et al. (2021) proposed the

staff of long-term care institutions should take into account the implementation of education and

training and fire safety equipment when implementing emergency response to fires[8].

In References, added 7.       Tsai, S.Y. Study on Case Analysis and Key Factors of Fire Prevention and

Evacuation Safety Improvement in Residential Long-term Care Institutions.China University of Science

and Technology. Taipei Taiwan 2021. 8.  Huang Y. H. ; Chen Y.F. ; Ni S. C. The Impact of Fire and Smoke

After the Disaster in the Long Term Care Facilities. Combustion Quarterly, 2021, 113, 28-42.

Point 4: Correction 2. 4 Data Analysis.

Response 4: Correction 2. 4 Data Analysis to Data Collection and Data Analysis.

Point 5: The average age of the supplementary experimental and control groups.

Response 5: In 3.1. Participants’ Demographics, added the average of the experimental group was 35.6 years old, and the average of the control group was 41.9 years old.
